# Clavicular Malignancies: A Borderline Surgical Management

**DOI:** 10.3390/medicina58070910

**Published:** 2022-07-08

**Authors:** Claudiu-Eduard Nistor, Adrian Ciuche, Anca-Pati Cucu, Bogdan Serban, Adrian Cursaru, Bogdan Cretu, Catalin Cirstoiu

**Affiliations:** 1Department of Thoracic Surgery, Central Military Emergency University Hospital Bucharest, University of Medicine and Pharmacy “Carol Davila”, 010825 Bucharest, Romania; ncd5879@gmail.com (C.-E.N.); adi_ciuche@yahoo.com (A.C.); ancutapati@gmail.com (A.-P.C.); 2Department Orthopedics and Traumatology, University of Medicine and Pharmacy “Carol Davila”, University Emergency Hospital, 050098 Bucharest, Romania; serbann.bogdan@yahoo.com (B.S.); cursaru_adrian@yahoo.com (A.C.); cirstoiu_catalin@yahoo.com (C.C.)

**Keywords:** clavicular malignancies, clavicular resection and reconstruction, clavicular metastasis, complex thoracic tumors, claviculectomy

## Abstract

Nearly 1% of all bone cancers are primary clavicular tumors and because of their rarity, treating clinicians are unfamiliar with their diagnosis, classification, treatment options, and prognosis. In terms of preserving function and avoiding complications, clavicle reconstruction seems logical; however, further studies are needed to support this measure. Reconstruction techniques are difficult taking into account the anatomical structures surrounding the clavicle. When chest wall defects are present, a multidisciplinary team, including an orthopedist and thoracic and plastic surgeons, is of paramount importance for optimal surgical management. Malignant clavicle tumors may include primary and secondary malignancies and neighboring tumors with clavicular invasion. Surgical resection of complex thoracic tumors invading the clavicles can result in larger defects, requiring chest wall reconstruction, which is a substantial challenge for surgeons. Correct diagnosis with proper preoperative planning is essential for limiting complications. Post-resection reconstruction of the partial or total claviculectomy is important for several reasons, including maintaining the biomechanics of the scapular girdle, protecting the vessels and nerves, reducing pain, and maintaining the anatomical appearance of the shoulder. The chest wall resection and reconstruction techniques can involve either partial or full chest wall thickness, influencing the choice of reconstructive technique and materials. In the present paper, we aimed to synthesize the anatomical and physiopathological aspects and the small number of therapeutic surgical options that are currently available for these patients.

## 1. Introduction

Primary clavicular tumors constitute less than 1% of all bone tumors. Due to this rarity, treating physicians have limited experience with their diagnosis, classification, treatment options, and prognosis [1].

Malignant clavicle tumors may include primary and secondary malignancies and neighboring tumors with clavicular invasion. The importance of this pathology derives from the particular anatomical location of the clavicle, relating to structures of vital importance, as well as the limited reconstructive possibilities after surgical treatment. For primary or single secondary tumors, the optimal treatment is complete or partial resection, with or without subsequent reconstruction. A reported series of cases exhibit the presence of tumors in both the acromial end and diaphyseal region [2,3,4]; however, there is not yet any standardized algorithm for their management. Notably, the clavicle can be considered a “special” type of bone from the perspectives of its embryology, morphology, localization, and function [2,3,5].

In the present paper, we aimed to synthesize the anatomical and physiopathological aspects and the small number of therapeutic surgical options that are currently available for these patients; we reviewed a total of 51 papers and book chapters and by presenting four different malignancies of the clavicle and manubrium.

## 2. The Clavicle: A “Special Bone”

The clavicle, or collar bone, is a long (6 inches in adults), tubular, S-shaped bone that is subcutaneous throughout and can be considered “special” for several reasons. It is the first bone to ossify in the embryo, with its two first ossification centers developing between the fifth and sixth weeks of gestation. It is also the last bone to fully ossify at the age of 25–26 years. Unlike other long bones, the clavicle develops via intramembranous ossification [2,3,6,7,8,9,10,11,12].

The clavicle is the only long bone in the human body that is positioned horizontally. It acts as a link between the upper limbs and the thorax, connecting the axial to the appendicular skeleton. Although still under debate, Inman and Saunders propose that the clavicle serves three primary functions: protection of the underlying neurovascular structures, a bony framework for the attachment of muscles, and strut support for the scapula [13].

The structure of the clavicle differs from other long bones in that it almost lacks a medullary cavity and comprises dense trabecular bone with a reduced vascular supply. It offers attachment to several muscles, including the deltoid, trapezius, and pectoralis muscles, and thereby contributes to the upper extremity movement. It also harbors paramount cervicothoracic structures, including the brachial plexus, the subclavian artery, and the pulmonary apex [7,9,11].

## 3. Clavicular Tumors: Diagnosis

Malignant clavicular tumors present with local pain, a palpable firm mass or pathological bone fracture, or with the clinical symptoms of the primary tumor (in cases of metastasis). Clavicular chondrosarcomas may present with skin ulceration, swelling, or, in rare cases, Horner’s syndrome or even thoracic outlet syndrome [14,15]. Surgical treatment is performed with the dual aims of removing tumoral tissue and alleviating symptoms.

While bone tumors can arise spontaneously, a substantial number occur in the context of a hereditary disorder, thus necessitating a detailed family history for every new case. Some of the most common primitive tumors in the clavicle include eosinophilic granuloma and tumors secondary to hematological disease [16]. Plasmacytoma, osteosarcoma, Ewings sarcoma, and post-radiotherapy sarcomas may also occur in the collarbone [17]. Although the literature describes a multitude of malignant lesions that may exist in the clavicle, this is not a common occurrence (Figure 1) [18,19,20,21,22,23,24].

Tumors originating in the lung, anterior chest wall (bones/soft tissue), breast, anterior mediastinum, or thyroid can invade the clavicle. Notably, the skeleton is the third site for metastasis, after the lung and liver [25]. In all of these cases, the management algorithm considers the tumor’s origin, the structures invaded, and the size of the chest wall defect after tumor resection with respect to the oncological principle of free margins [26].

Given the proximity of the clavicle to major vessels, nerves, and the thorax, there is a need for standard imaging—including plain film, computed tomography, and magnetic resonance imaging—to evaluate bone destruction and invasion of surrounding tissues (muscles, vessels, and nerves). Whole-body scintigraphy using technetium-99m methylene diphosphonate (99 mTc MDP) and positron emission tomography scan (PET scan) studies are required to diagnose metastases in the bones or other structures. Importantly, the high variability of the radiological appearance complicates the diagnosis of these masses alone [17].

Chondrosarcoma is the second most common bone cancer, with 20% occurring in the shoulder girdle, up to 1% being metastasis from other cancers, and only rare occurrences in the clavicular location (Figure 2) [27]. On the other hand, one-third of chest wall tumors are chondrosarcomas, making this the most common primary bone tumor of the chest wall [17]. Clavicle chondrosarcoma exhibits a lesion-like appearance, similar to that found in other skeletal regions, with specific intratumoral calcifications present in bulky soft tissue masses [28]. Clavicular osteosarcoma is not associated with the classic periosteal reaction or with the mineralized matrix, making imaging-based diagnosis difficult.

Laboratory examinations are relatively unimportant for diagnosing clavicular tumors, except in cases of metastasis, where screening for specific tumor markers might indicate the primary tumor. Preoperative staging is followed by a biopsy (fine needle/excisional or incisional) to confirm whether the tumor is primary or secondary, thus establishing the final diagnosis.

The purpose of surgical treatment is the resection of the tumoral mass, followed by immediate reconstruction if stability must be restored. Since the biological healing process is no longer reliable, various biological or synthetic materials must be used. Primary tumors of the clavicle without locoregional invasion are suitable for complete or partial resection within oncological limits, with or without subsequent reconstruction. Secondary tumors should be managed according to the tumor origin and the number of secondary determinations. When facing a single secondary tumor of the clavicle, the optimal approach is oncological resection with or without subsequent reconstruction.

## 4. Surgical Management and Reconstruction of Clavicular Malignancies

In cases requiring reconstruction, several aspects must be considered: what structure is missing, what is left, what should be replaced, and what are the available means of reconstruction. Reconstruction is influenced by the location, size, and depth of the defect, the viability and quality of adjacent structures (previous radiotherapy, presence of scars), and previous surgery in the area [26]. Post-resection reconstructive methods are limited in cases of unique primary or secondary clavicular tumors. The external third resection of the clavicle does not require reconstruction as long as the coracoclavicular ligaments remain attached but, of course, such cases are rare in discussions of malignancies.

Partial or complete claviculectomy involves substantial peri-operative risks, such as lesions of the subclavian plexus or the brachial plexus, as well as infectious risks [29,30,31]. Post-resection reconstruction of the partial or total claviculectomy is important for several reasons—including maintaining the biomechanics of the scapular girdle, protecting the vessels and nerves, reducing pain, and maintaining the anatomical appearance of the shoulder. One reconstructive method following a complete resection of the clavicle is with a vascularized autograft fibula or rib, and several case reports describe the surgical technique but without comparing claviculectomy with reconstructive procedures [32,33,34]. Alternative reconstruction options involve acrylic cement reinforced with plate and screws or with Kirschner brooches, which may avoid possible complications secondary to complete resection of the collarbone (Figure 2). After reviewing five cases treated with this procedure, the authors concluded that bone cement prosthesis for bone defect reconstruction after tumor resection can maintain the contour of the shoulder and reduce the complications ascribed to the claviculectomy, and the procedure is effective and feasible [35]. Moreover, the “Oklahoma prosthesis” reconstructive method, with a cement allograft, has been successfully used to maintain the biomechanical integrity of the scapular girdle in cases of clavicular middle third secondary tumors that invade the sternum and rib. The technique consists of an en bloc clavilculectomy and chest wall resection with a method of reconstruction using a single methyl methacrylate and prolene composite prosthesis in a configuration resembling the state of Oklahoma [36].

Due to the small number of studies analyzing clavicular malignancies treated by resection and reconstruction, and the fact that the vast majority of these studies are case reports or include only a small number of cases, it cannot be firmly stated that reconstruction is the best option in terms of preserving scapular girdle functionality and long-term survival rates. To state that reconstruction is mandatory after claviculectomy, there is a need for studies focused on specific patient groups, with analyses of functionality over time and long-term patient survival.

## 5. Surgical Management and Reconstruction of Complex Thoracic Tumors Invading the Clavicles

Surgical resection of complex thoracic tumors invading the clavicles can result in larger defects, requiring chest wall reconstruction, which is a substantial challenge for surgeons. Reconstruction is necessary for the restoration of the bony framework, prevention of respiratory mechanics impairment, protection of the underlying structures, preventing herniation of the thoracic viscera, and minimization of deformity. If tumors involve the anterior chest wall, the surgical management is complex. The chest wall resection and reconstruction techniques can involve either partial or full chest wall thickness, influencing the choice of reconstructive technique and materials (Figure 3).

In the case of a primary chest wall tumor, a 4 to 5 cm tumor-free margin must be achieved, and the immediate adjacent ribs should also be resected [37,38]. For metastasis, the rule described by Leonardi requires a 2 cm parietectomy away from the tumor. Sternal manubrium tumors require resection of the internal third of the clavicles [39]. Reconstruction is not required following manubrial resection, en bloc or not, with either one or both clavicles [37,38]. In a case series of manubrial malignant tumors, reconstruction was successfully performed with autogenous rib grafts [40]. However, if the tumor affects at least 3 or 4 ribs, a paradoxical rib cage might occur, preventing complete re-expansion of the contralateral lung after surgery [37,38,41,42,43,44,45].

There are currently no guidelines regarding the indications and limits for anterior chest wall reconstruction; however, most studies agree on the necessity of reconstruction for defects larger than 5 cm or including at least four ribs [43,45,46]. Anterior chest wall defects require more than a single layer of material for reconstruction—usually a mixture of flaps, mesh, and prosthetic material for bone replacement. There are two types of chest wall reconstruction that can be used, alone or together: rigid reconstruction, which mainly serves to stabilize the thoracic cage and restore function, and soft tissue/musculocutaneous reconstruction, which practically covers the lack of substance and addresses the aesthetic aspect. The use of plastic surgery techniques for skin approximation can be quite useful for a moderate cutaneous defect. For a larger defect, various flaps can be used by rotation, transposition, or interposition. Rigid reconstruction re-establishes the chest wall integrity in cases requiring full-thickness reconstruction.

An ideal prosthetic material should be readily available, sufficiently rigid to abolish paradoxical chest movements but also malleable, biologically inert, sterile, germ-resistant, and radiolucent [47]. Recently, titanium meshes have been used with promising results, showing enhanced resistance compared to synthetic meshes, as well as good plasticity and adaptability [48]. Synthetic materials used for reconstruction include polypropylene, polyglactin, methylmethacrylate, polytetrafluoroethylene, and nylon meshes. A wide variety of bioprosthetic materials are also available, including cadaveric human dermis, porcine dermis, bovine pericardium, bovine dermis, and acellular collagen matrix meshes. Cryopreserved homografts can be selected based on availability, type or size of the defect, and the patient’s habitus, age, and associated conditions.

Rigid chest wall reconstruction can be performed using both biological and synthetic materials. Biological materials have the advantage of biocompatibility but are less resistant over time, increase the operating time, and can result in donor site complications. On the other hand, synthetic materials have an increased resistance but might lead to infection, extensive fibrosis, migration, or shredding/rupture. Osteosynthesis systems, such as Stratos or MatrixRIB fixation, have proven utility for rigid reconstruction (Figure 4). Customized titanium implants manufactured using 3D prototyping technology, combined with a myocutaneous pedicled flap, have been proven safe for use in extended anterior chest wall resection [49,50].

## 6. Conclusions

Malignant tumors of the clavicle are rare, and their prognosis is uncertain. Claviculectomy is the main curative treatment in cases of primary or unique secondary tumors. There are several possible approaches to post-resection reconstruction, and the approach must be selected with consideration of the tumor type, in terms of prognosis, as well as the patient’s age and activity level. Claviculectomy alone or followed by reconstruction with vascularized autograft fibula or rib, bone cement prosthesis, or the “Oklahoma prosthesis“ are the main options in treating this pathology. The choice of technique will be made taking into account whether the tumor is primary or secondary, its local extension and, of course, the patient’s age and needs, post-resection reconstruction should be performed in cases where thoracic limb function and local aesthetics are necessary to be preserved. When chest wall defects are present, a multidisciplinary team, including orthopedic, thoracic, and plastic surgeons, is of paramount importance for optimal surgical management. In terms of preserving function and avoiding complications, clavicle reconstruction seems logical; however, further studies are needed to support this measure. In cases with anterior chest wall defects, reconstruction (either soft tissue or rigid) is typically recommended. The variety of materials available for reconstruction, and the potential use of 3D custom-designed implants, enable more accurate reconstruction in cases with complex associated defects.

However, reconstruction is only relevant for total clavicle resection and subclavicle resection, as the size and location of the clavicle tumor may differ and the extent of tumor resection may be different.

## Figures and Tables

**Figure 1 medicina-58-00910-f001:**
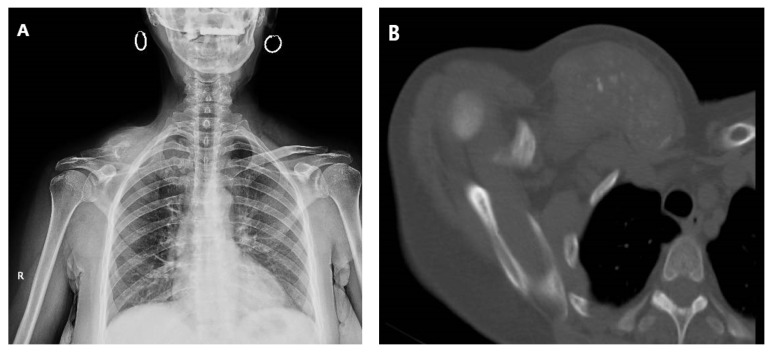
Plain radiograph (**A**) and tomography (**B**) revealing extensive locoregional clavicle metastasis in a 60-year-old female, secondary to cervix adenocarcinoma. The patient presented with multiple metastases at diagnosis.

**Figure 2 medicina-58-00910-f002:**
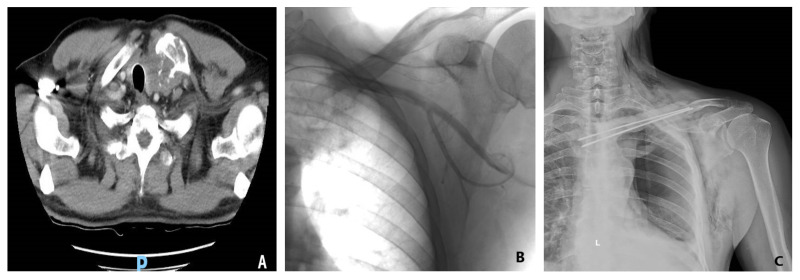
(**A**) Preoperative tomography of a 66-year-old male diagnosed with clavicle condrosarcoma. (**B**) Preoperative angiography and embolization of the tumor were performed prior to resection to limit intraoperative blood loss. (**C**) Postoperative image of acrylic cement reinforced with Kirschner brooches reconstruction used after tumoral resection.

**Figure 3 medicina-58-00910-f003:**
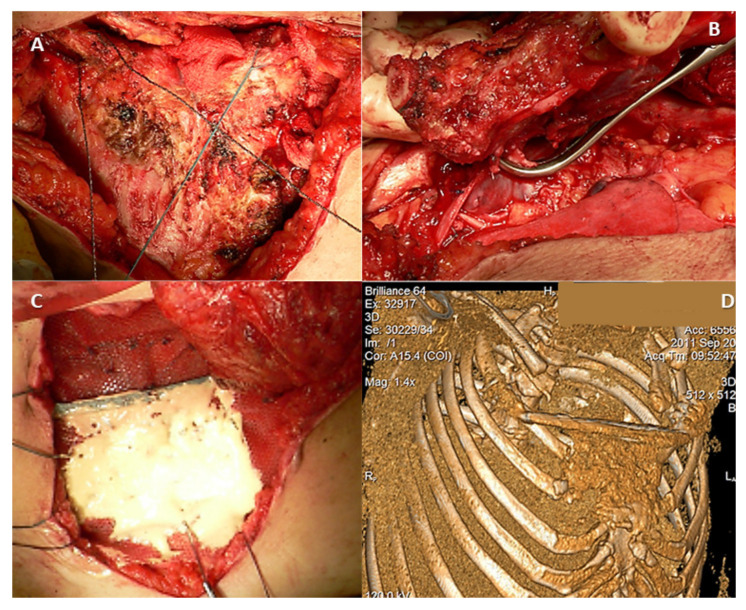
Intraoperative aspect of a 64-year-old patient diagnosed with a manubrial tumor. A Gigli wire saw (**A**) was used for en bloc bilateral resection of the medial third of the clavicles, the cartilaginous portion of the first rib, and the manubrium (**B**). (**C**) Polypropylene mesh and Kryptonite cementum were used for rigid anterior chest wall reconstruction. (**D**) A 3D-CT scan reconstruction at three months after surgery shows the neo-sternum.

**Figure 4 medicina-58-00910-f004:**
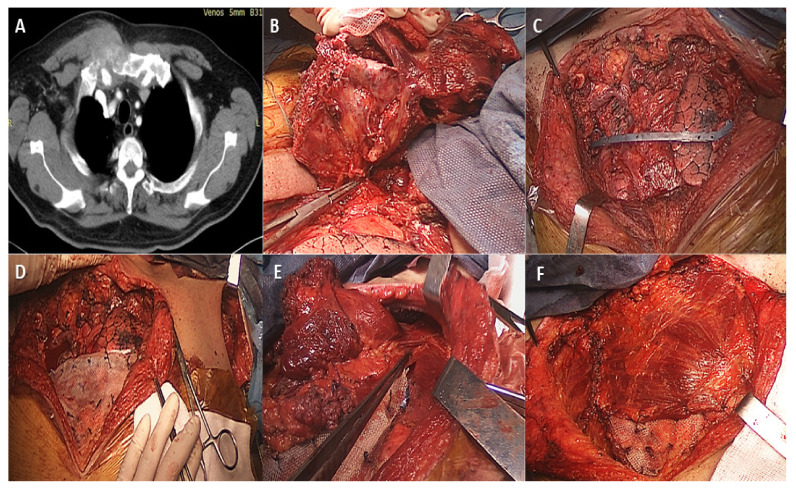
(**A**) Preoperative CT scan of a 55-year-old patient diagnosed with a manubrial tumor. (**B**) A large resection of the tumor was performed, within oncological limits. (**C**) After resection, rigid reconstruction was performed using a Stratos titanium bar on the upper side of the operative field. The medial part of the clavicle is shown. (**D**) Next, a polypropylene mesh was utilized in the left upper corner. The incision used for sectioning of the deltoid tuberosity insertion of the left pectoralis major is shown. (**E**) For soft-tissue reconstruction, a pectoralis major flap was harvested (main pedicle at the tip of the clamp). (**F**) The intraoperative aspect with the pectoralis major flap was secured over the polypropylene mesh.

## Data Availability

Further data concerning the study can be obtained by contacting the corresponding author.

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
