# Peer review of "Clavicular Malignancies: A Borderline Surgical Management"

_medicina, 2022, doi:10.3390/medicina58070910_

Round 1

Reviewer 1 Report

It is an interesting work that could provide interesting insights on clavicular tumors. However, it can be further improved by addressing a few minor points listed below.

1.      Some type mistakes should be revised. For example, Line 217: 55yearold

2.      It is not clear how many samples were used for the study, please clarify in the figure legend or main text.

Author Response

Thank you for your comments.

We have revised the points listed by you.

  1. "Some type mistakes should be revised. For example, Line 217: 55yearold", we revised it  
  2. "It is not clear how many samples were used for the study, please clarify in the figure legend or main text.", we have stated in the main text how many papers and how many cases were discussed (line 46 and 47).

Thank you.

Reviewer 2 Report

In this article, the authors tried to provide an overview of management of the clavicular malignancies. Diagnosis and treatment of this disease are challenging to the clinical practitioners given the rarity of the disease and the difficulty of the surgical treatment.

Because of the rarity of the clavicular malignancies, it is impossible to draw a firm conclusion regarding the standard management of this disease, the relevant articles are usually case reports and case series. However, it is possible to present the typical outcomes of different surgical strategies regarding the clavicular malignancies, to my knowledge, the necessity of reconstruction of the clavicle is controversial since some reports showed favorable functionality after claviculectomy without reconstruction, the authors may present all these outcomes in the form of a table.

Surgical management of the clavicular malignancies may be described in more details, for example, by showing objective data and outcomes of typical articles. Comparisons regarding the different surgical methods is necessary.

Because the scale of resection of the tumor may be varied due to different size and location of the clavicular tumor, but reconstruction is only of importance for cases having total and subtotal claviculectomy, this content may be put more explicitly.

It may be better if the authors make more comparisons of outcomes of different papers in the discussion section.

I believe there is more to be demonstrated about the surgical treatment of the clavicular malignancies, therefore I recommend major revision of this paper.

Author Response

Thank you very much for you comments.

  1. "the necessity of reconstruction of the clavicle is controversial since some reports showed favorable functionality after claviculectomy without reconstruction" we have stated that external third does not require reconstruction (line 124-126). We have presented the benefits of reconstruction after total or subtotal claviculectomy without stating that reconstruction is mandatory (line 147-153), furthermore we have underlined the need for studies comparing resection with and without reconstruction for such a statement to be made (line 147-153). Thank you

2. "Surgical management of the clavicular malignancies may be described in more details" , we have addressed this recommendation. Thank you.

3. "Because the scale of resection of the tumor may be varied due to different size and location of the clavicular tumor, but reconstruction is only of importance for cases having total and subtotal claviculectomy, this content may be put more explicitly." , we have reiterated in the main text this statement (line 220-222). Thank you.

4. "It may be better if the authors make more comparisons of outcomes of different papers in the discussion section.". Thank you for you comment, we addressed this recommendation.